Forelimb musculoskeletal-tendinous growth in frogs

Soliz Mónica monica.c.soliz@gmail.com 1
Tulli María Jose majotulli@gmail.com 2 3
Abdala Virginia 4
1 Cátedra Vertebrados, Facultad de Ciencias Naturales, Universidad Nacional de Salta , Salta , Argentina
2 Unidad Ejecutora Lillo (CONICET-FML), Cátedra de Biología Animal, Facultad de Ciencias Naturales (UNT) , Tucumán , Argentina
3 Cátedra de Biología Animal, Facultad de Ciencias Naturales, Universidad Nacional de Tucumán , Tucuman , Argentina
4 Instituto de Biodiversidad Neotropical (IBN), Cátedra de Biología General, Facultad de Ciencias Naturales, UNT, UNT-CONICET , Tucumán , Argentina
Robillard Tony
Electronic publication date: 2020 Feb 25
Publication date: 2020
Volume: 8
Electronic Location ID: e8618
Received 2019 Sep 12; Accepted 2020 Jan 22
Copyright: ©2020 Soliz et al.
Copyright year: 2020
Copyright holder: Soliz et al.
License: This is an open access article distributed under the terms of the Creative Commons Attribution License, which permits unrestricted use, distribution, reproduction and adaptation in any medium and for any purpose provided that it is properly attributed. For attribution, the original author(s), title, publication source (PeerJ) and either DOI or URL of the article must be cited.
License URL: https://creativecommons.org/licenses/by/4.0/

Keywords: Muscles, Bones, Tendons, Allometry, Ontogeny, Anuran

Funding: PIUNT 26/G625 PIP CONICET 389 PICT 2016-2772 This work was funded by the PIUNT 26/G625 and PIP CONICET 389 and PICT 2016-2772. The funders had no role in study design, data collection and analysis, decision to publish, or preparation of the manuscript.

==============================
The tendons unite and transmit the strength of the muscles to the bones, allowing movement dexterity, the distribution of the strength of the limbs to the digits, and an improved muscle performance for a wide range of locomotor activities. Tissue differentiation and maturation of the structures involved in locomotion are completed during the juvenile stage; however, few studies have investigated the ontogenetic variation of the musculoskeletal-tendinous system. We ask whether all those integrated tissues and limb structures growth synchronically between them and along with body length. We examined the ontogenetic variation in selected muscles, tendons and bones of the forelimbs in seventy-seven specimens belonging to seven anuran species of different clades and of three age categories, and investigate the relative growth of the forelimb musculoskeletal-tendinous structures throughout ontogeny. Ten muscles and nine tendons and their respective large bones (humerus and radioulna) were removed intact, and their length was measured and analyzed through a multivariate approach of allometry. We obtained an allometry coefficient, which indicates how the coefficient departures from isometry as well as allometric trends. Our data suggest that along with the post-metamorphic ontogeny, muscles tend to elongate proportionally to bone length, with a positive allometric trend. On the contrary, tendons show a negative allometric growth trend. Only two species show different patterns: Rhinella granulosa and Physalaemus biligonigerus, with an isometric and positive growth of muscles and bones, and most tendons being isometric.

Introduction

The musculotendinous system is particularly active in the general limb movements. The tendons unite and transmit the strength of the muscles to the bones, allowing movement dexterity, and the distribution of the strength of the limbs to the digits (Kardong, 2002). The release of the elastic energy of muscular aponeuroses and ligaments amplifies the power and reduces muscle work (Roberts, 2002; Biewener, 2003). The capacity for the differential jump between frog species is related to the relative amount of musculature of the hindlimb and the use of the energy stored in tendons and ligaments (Emerson, 1978). Further, tendons improve muscle performance for a wide range of locomotor activities (Roberts, 2002).

During ontogeny, the characteristics of muscle architecture and connective tissue vary according to the body length increase, and to the increase of daily activity functional demands. Thus, many of the evolutionary and developmental transformations in structures related to the locomotor function in anurans occur during larval stages and metamorphosis (Ročkova & Roček, 2005; Púgener & Maglia, 2009; Manzano et al., 2013; Fabrezi et al., 2014). Studies on anuran ontogeny reveal that locomotor modes (walking, jumping, and swimming) are achieved before the acquisition of the pelvis-sacral-urostil complex coordination, and the hindlimbs are acting as a unit (Fabrezi et al., 2014). Although metamorphosis has been considered as the period in which the most critical anatomical characteristics for adult locomotion are developed (for example, those related to girdles and limbs; Fabrezi et al., 2014), it is known that tissue differentiation and maturation of the structures involved in locomotion are completed during the juvenile stage (Vera, Ponssa & Abdala, 2015).

There are numerous studies on anuran ontogeny (Hanken & Hall, 1984; De Sá, 1988; Hall & Larsen Jr, 1998; Banbury & Maglia, 2006; Handrigan & Wassersug, 2007; Handrigan, Haas & Wassersug, 2007; Manzano et al., 2013 among many others); and on the relationship between different ontogenetic trajectories and modes of life (Haas, 1999; Haas & Richards, 1998; Ročková & Roček, 2005; Púgener & Maglia, 2009; Manzano et al., 2013; Fabrezi et al., 2014; Soliz & Ponssa, 2016). However, considerably less attention has been given to the changes occurring through juveniles and metamorphic stages (Ponssa & Vera Candioti, 2012). Juveniles of most species must maneuver in the same environment and avoid the same predators as adults, but they undergo ontogenetic changes that allow them to sprint and jump almost as fast and far as adults. The allometric changes in juveniles include relatively longer limbs, muscular forces, and relatively greater contractile speeds and higher muscular mechanical advantage (Carrier, 1995; Ponssa & Vera Candioti, 2012).

Several researchers found that subtle structural differences in the proportion of hindlimbs would facilitate functional diversity, allowing for wide-ranging exploitation of environments (Duellman & Trueb, 1994; Emerson, 1978; Emerson, 1979; Emerson, 1983; Emerson, 1985; Emerson & De Jongh, 1980; Nauwelaerts, Ramsay & Aerts, 2007). Ontogenetic processes have been considered responsible of the morphological variability among species and could have a profound impact on the shape of morphological structures (Thompson, 1942; Gould, 1977; Alberch, Gould & Wake, 1979; Calder III, 1984; Schmidt-Nielsen, 1984; Emerson & Bramble, 1993; Reilly, Wiley & Meinhardt, 1997; Vukov et al., 2018). However, few studies have investigated the ontogenetic variation of the musculotendinous system, which could be responsible for generating and transmitting force to produce and control body movements. This active force is produced by the muscle fibers and is transmitted to the bones through aponeuroses and tendons (Biewener, 1998; Böl, Leichsenring & Siebert, 2017).

Here, we examined the ontogenetic variation in selected muscles, tendons, and bones of the forelimbs in seven anuran species. Our main goal was to investigate the relative growth of the forelimb musculoskeletal-tendinous structures throughout ontogeny. All the ontogenetic stages considered here belong to the fully functional category (Muntz, 1976), implying that all tissues and limb structures are integrated. We ask whether all those integrated tissues and limb structures growth synchronically between them and along with body length, and hypothesize that our data will show a general allometric pattern of increased growth rate with larger body length. The study of the effect of the ontogeny onto the longitudinal growth of the musculoskeletal-tendinous structures of frogs represents a starting point to a more global analysis considering other variables such as volume or cross-sectional areas.

Material and Methods

The right forelimb of seventy-seven specimens belonging to six species encompassing different clades of the anuran phylogeny (Duellman & Trueb, 1994 and Pyron & Wiens, 2011) were dissected (Table 1). Then, 10 muscles and 9 tendons, and their respective large bones (humerus and radioulna) (Table 2) were removed intact, and their length was measured (Fig. 1). The specimens were staged in three estimated categories according to Gosner (1960): metamorphic 46 (2); juvenil (3); and adult (4). Dissections of the anatomical traits were performed between the origin and insertion points under a binocular microscope (Nikon SMZ645), and were measured in mm with digital callipers (±0.01 mm; Mitutoyo CD-15B; Mitutoyo Corp., Kure, Japan). When referring to muscles, abbreviation “m.” before muscle names was added; otherwise names refer to tendons associated with those muscles (Table 1). The terminology used follows Prikryl et al. (2009), Abdala & Diogo (2010) and Diogo & Ziermann (2014). Data of muscles measurements, body length and number of individuals used per species are detailed as Table S1. All the examined specimens are deposited in systematic collections, and listed in Table 1.

Table 1 List of anuran specimens examined.

Acronyms for Argentinian institutions where specimens are deposited and personal catalogs: MCN, Museo de Ciencias Naturales de Salta, FML, Fundacin Miguel Lillo; MS, field number of Mnica Soliz to be deposited at FML.

Species	N	Specimen number	
Rhinella arenarum	18	MCN594, MCN849, MCN858, MCN579, MCN776	
Rhinella granulosa	9	MCN775, MCN994, MCN779, MS0130, MS0131, MS0132	
Pleurodema borellii	29	MCN707, MCN592, MCN844, MS0133, MS0134, MS0135	
Physalaemus biligonigerus	11	MCN731, MS0137, MS136	
Leptodactylus chaquensis	12	MCN327, MCN166, MCN327, MCN583, MCN789	
Trachycephalus typhonius	11	FML29062, FML29133, FML29167, FML29168, MS0129	

Table 2 List of abbreviations of forelimb muscle, tendons, bones and body length of measured specimens.

Muscles	Abbreviations	Tendon	Abbreviations	Group	
Extensor					
Triceps scapularis medialis	SMT	x	SMTT	Triceps	
Triceps humeralis lateralis	Hlat	x	HlatT	Triceps	
Triceps humeralis medialis	Hmed	x	HmedT	Triceps	
Extensor carpi radialis	Ecr	x	EcrT	Extensor of the forearm	
Extensor carpi ulnaris	Ecul	x	EculT	Extensor of the forearm	
Extensor digitorum	EDig			Extensor of the forearm	
					
Flexor					
Coracoradialis	C	x	CT	Flexor of the arm	
Flexor carpi radialis	Fcr	x	FcrT	Flexor of the forearm	
Flexor carpi ulnaris	Fcul	x	FculT	Flexor of the forearm	
Flexor digitorum comunis	Fdc	x	FdcT	Flexor of the forearm	
Bones					
Humerus length	HL			Arm	
Radioulna length	RUL			Forearm	
Bodylength	BL				

Figure 1 Muscles and tendons of the right forelimb of anuran species, and their respective large bones.

(A) Dorsal view. (B) Ventral view. Light grey muscles and dark gray tendons. (C) Coracoradialis. CT, Coracoradialis tendon; Ecr, Extensor carpi radialis; EcrT, Extensor carpi radialis tendon; Ecul, Extensor carpi ulnaris; EculT: Extensor carpi ulnaris tendon; EDig, Extensor digitorum; Fcr, Flexor carpi radialis FcrT Flexor carpi radialis tendon; Fcul, Flexor carpi ulnaris; FculT, Flexor carpi ulnaris tendón, Fdc, Flexor digitorum comunis; FdcT, Flexor digitorum comunis tendon; H, humerus; Hlat, Triceps humeralis lateralis; HlatT, Triceps humeralis lateralis tendon; Hmed, Triceps humeralis medialis; HmedT, Triceps humeralis medialis tendon; RU, radioulna; SMT, Triceps scapularis medialis; SMTT, Triceps scapularis medialis tendon. Scale bars: five mm.

Statistical analysis

To estimate the scaling of muscles and tendons throughout postnatal ontogeny of the forelimb we performed a multivariate allometric tests based on the generalized allometric equation proposed by Jolicoeur (1963). We performed a principal component analysis (PCA) to obtain the 1st PC eigenvector that expresses the scaling relationships among all variables with the latent size regarded as a latent variable affecting all measured variables simultaneously (Giannini, Abdala & Flores, 2004; Giannini et al., 2010). This eigenvector is extracted from a variance–covariance matrix of log10-transformed variables and scaled to unity (Jolicoeur, 1963). The significance of multivariate allometry coefficients was tested using a resampling strategy based on jackknife. Each specimen was removed from the sample at a time, generating n pseudovalues to calculate confidence intervals (CIs) for the original coefficients (Giannini, Abdala & Flores, 2004; Flores, Giannini & Abdala, 2006). If the interval excluded an expected value of isometry, the variable was considered positively or negatively allometric. For all multivariate coefficients of allometry, the expected value of isometry, which depends only on the number of variables (p), is calculated as 1/p0.5 (0.21) for our set of 22 variables). Trimming the largest and smallest m pseudovalues (with m = 1) for each variable may significantly decrease the standard deviations calculated under jack-knife, and allow for more accurate allometric estimations; Giannini, Abdala & Flores, 2004). Here, untrimmed and trimmed calculations are reported, but the chosen results are those with either lower average standard deviation, or lower bias (with the latter defined as the difference between observed and jackknifed allometry coefficient; (Giannini, Abdala & Flores, 2004).

Results

Individual values of the analysed morphological variables are shown in Table S1. Scaling analyses describing ontogenetic growth in length and width of forelimb muscles and tendons in seven frog species are shown in Table 3. Results of allometry multivariate analyses are given in Table 3. We report untrimmed (m = 0) as well as trimmed (m = 1) calculations of confidence intervals, opting for the results with lower average standard deviation or bias, trimmed with CI 95% which combines the conservative safety of interval and estimated bias in these analyses .

Table 3 Summary of allometric trends in the seven species of anuran for 20 variables investigated.

The used symbols are: “+” (accelerated with respect to overall size or positive allometric), “=” (respect to overall size or negative allometric), “=” (isometric).

Species	Rhinella arenarum	Rhinella granulosa	Trachycephalustyphonius	Physalaemus biligonigerus	Leptodactylus chaquensis	Pleurodema borelli	
Variables	Untrimmed	Trimmed	Untrimmed	Trimmed	Untrimmed	Trimmed	Untrimmed	Trimmed	Untrimmed	Trimmed	Untrimmed	Trimmed	
BL	+	+	=	=	=	=	+	=	+	+	=	=	
HL	=	=	=	=	+	=	+	+	=	=	=	=	
RUL	=	+	=	=	=	=	+	+	+	=	+	+	
SM	=	+	=	=	+	+	+	=	+	=	=	=	
SMTL	=	+	–	=	=	=	=	=	+	=	–	=	
Hlat	+	+	=	=	+	=	=	=	+	+	+	+	
HlatTL	=	−	–	–	=	=	=	=	–	=	–	=	
Hmed	+	+	=	=	=	=	–	=	+	+	=	=	
HmedTL	=	−	=	=	–	–	–	–	–	=	=	=	
Edig	+	+	=	=	+	+	+	=	+	+	+	+	
Ecul	+	+	=	=	+	=	=	=	+	+	+	+	
EculT	=	=	=	=	–	=	–	–	=	=	–	–	
Ecr	=	=	=	=	+	=	–	=	=	=	=	=	
EcrT	=	+	+	=	–	=	–	–	+	=	=	=	
C	=	−	–	=	=	=	+	=	–	=	–	=	
CTL	=	=	=	=	=	=	+	+	=	=	=	=	
Fdc	+	+	=	=	=	=	=	=	+	+	=	=	
FdcT	−	−	=	=	–	=	=	=	–	–	–	=	
Fcul	=	+	+	=	=	=	=	=	+	=	+	=	
FculT	−	−	=	=	=	=	=	=	–	–	=	=	
Fcr	+	+	+	=	=	=	=	=	+	+	=	=	
FcrT	−	−	=	=	=	=	+	=	–	–	=	=	

Musculoskeletal-tendinous allometry in Rhinella arenarum

Trends obtained with untrimmed and trimmed values differed only in relation to five variables: radioulna length (RUL), extensor carpi radialis tendon length (EcrT), humeralis medialis tendon (HmedT) and m. coracoradialis (C). The greatest departure was observed in two variables, m. flexor digitorum communis (Fdc) and m. flexor carpi ulnaris (Fcul) lengths (−0.213 and −0.208, respectively). Two variables showed the smallest observed departure from isometry: m. flexor carpi radialis (FcrT) and flexor carpi ulnaris tendon lengths (FculT) (0.061 and −0.068, respectively). Based on this result, twelve variables significantly departed from isometry: radioulna length (RUL), m. humeralis lateralis (Hlat), m. extensor digitorum (Edig), m. extensor carpi ulnaris (Ecul), m. flexor digitorum communis (Fdc) and extensor carpi radialis tendon length (EcrT), and body length (BL), all positively allometric. M. coracoradialis (C), humeralis medialis tendon length (HmedT), flexor digitorum communis tendon length (FdcT), flexor carpi ulnaris tendon length (FculT) and flexor carpi radialis tendon length (FcrT) were all negatively allometric (Table S2).

Musculoskeletal-tendinous allometry in Rhinella granulosa

Trends obtained with untrimmed and trimmed values differed only in relation to five variables: scapularis medialis tendon length (SMT), extensor carpi radialis tendon length (EcrT), m. coracoradialis (C), m. flexor carpi ulnaris (Fcul) and m. flexor carpi radialis (Fcr). The greatest departure was observed in two variables, m. extensor carpi ulnaris (Ecul) and extensor carpi radialis tendon length (EcrT) (0.320 and 0.308, respectively). The smallest departure observed from isometry was found in scapularis medialis tendon length (SMT) and humeralis lateralis tendon length (HlatTL) (0.108 and 0.082, respectively). Based on this result, six variables significantly departed from isometry: humeralis lateralis tendon length (HlatT), scapularis medialis tendon length (SMT), and m. coracoradialis (C; all negatively allometric); and extensor carpi radialis tendon length (EcrT), m. flexor carpi ulnaris (Fcu) and m. flexor carpi radialis (Fcr; all positively allometric) (Table S3).

Musculoskeletal-tendinous allometry in Trachycephalus typhonius

Trends obtained with untrimmed and trimmed values differed only in relation to seven variables: humerus length (HL), humeralis lateralis (Hlat), m. extensor carpi radialis (Ecr) and tendon length (EcrT), m. extensor carpi ulnaris (Ecu) and tendon length (EcuT) and flexor digitorum communis tendon length (FdcT) (Table 3). The greatest departure was observed in three variables, m. scapularis medialis (SM) (0.272), extensor carpi radialis (Ecr) (0.259) and extensor digitorum communis (Edc) (0.265) (Table 3). The smallest observed departure from isometry was in extensor carpi ulnaris tendon length (EculT) and humeralis medialis tendon length (HmedT) (0.07 and 0.107, respectively). Based on this result, ten variables significantly departed from isometry: humeralis medialis tendon length (HmedT), extensor carpi radialis tendon length (EcrT), extensor carpi ulnaris tendon length (EcuT) and flexor digitorum communis tendon length (FdcT; all negatively allometric); and humerus length (HL), extensor carpi radialis tendon length (EcrT), m. flexor carpi ulnaris (Fcu) and m. flexor carpi radialis (Fcr; all positively allometric) (Table S4).

Musculoskeletal-tendinous allometry in Physalaemus biligonigerus

Trends obtained with untrimmed and trimmed values differed only in relation to seven variables: body length (BL), scapularis medialis (SM), humeralis medialis (Hlat), m. extensor digitorum (Edig), m. extensor carpi radialis (Ecr), m. coracoradialis (C) and flexor carpi radialis tendon length (FcrT) (Table 3). The greatest observed departure from isometry was observed radioulna length (RUL) and coracoradialis tendon length (CT) (0.278 and 0.277, respectively). The smallest departure was observed in three variables: extensor carpi ulnaris tendon length (EcuT) (0.069), extensor carpi radialis tendon length (EcrT) (0.082) and humeralis medialis tendon length (HmedT) (0.109). Based on this result, six variables significantly departed from isometry: humeralis medialis tendon length (HmedT), extensor carpi radialis tendon length (EcrT), extensor carpi ulnaris tendon length (EcuT; all negatively allometric), and humerus length (HL), radioulna length (RUL), coracoradialis tendon length (CT; all positively allometric) (Table S5).

Musculoskeletal-tendinous allometry in Leptodactylus chaquensis

Trends obtained with untrimmed and trimmed values differed only in relation to eight variables: radioluna (RUL), m. scapularis medialis (SM), scapularis medialis tendon length (SMT), humeralis medialis tendon length (HlatT), humeralis lateralis tendon length (HmedT), m. coracoradialis (C), extensor carpi radialis tendon length (EcrT), and flexor carpi ulnaris tendon length (FcuT) (Table 3E). The greatest observed departure from isometry was found in scapularis medialis tendon length (0.29), m. extensor carpi ulnaris (0.295) and extensor carpi radialis tendon length (0.315). The smallest departure was observed in four variables: humeralis medialis tendon length (0.09), flexor digitorum communis tendon length (0.02), flexor carpi ulnaris tendon length (EcuT) (0.068) and flexor carpi radialis tendon length (EcrT) (0.061). Based on this result, ten variables significantly departed from isometry: body length (BL), m. humeralis lateralis (Hlat), humeralis medialis (Hmed), m. extensor digitorum (Edig), m. extensor carpi ulnaris (Ecul), m. flexor digitorum communis (Fdc), m. flexor carpi radialis (Fcr; all positively allometric); and flexor carpi radialis tendon length (FcrT), flexor carpi ulnaris tendon length (FcuT), and flexor digitorum communis tendon length (FdT; all negatively allometric) (Table S6).

Musculoskeletal-tendinous allometry in Pleurodema borellii

Trends obtained with untrimmed and trimmed values differ only in relation to five variables: scapularis medialis tendon length (SMT), humeralis lateralis tendon length (HmedT), m. coracoradialis (C), flexor digitorum communis tendon length (FdcT) and m. flexor carpi ulnaris (Fcu) (Table 3). The greatest observed departure from isometry was observed in m. humeralis lateralis and m. extensor digitorum (0.262 and 0.301, respectively). The smallest departure was observed in two variables: extensor carpi ulnaris tendon length (EcuT) and flexor digitorum communis tendon length (FdcT) (0.05 and 0.11, respectively). Based on this result, five variables significantly departed from isometry: radioulna length, m. humeralis lateralis (Hlat), m. extensor digitorum (Edig), m. extensor carpi ulnaris (Ecul; all positively allometric), and m. extensor carpi ulnaris (Ecu; negatively allometric) (Table S7).

Discussion

The main goal of the present study was to investigate the relative growth patterns of the forelimb musculoskeletal-tendinous system in seven anuran species of three age categories. The general allometric growth patterns inferred from our data indicate that along with the post-metamorphic ontogeny of most studied anuran species, muscles tend to elongate proportionally to bone length, with a positive allometric trend. On the contrary, tendons show a negative allometric growth trend. Only two species show different trends: Rhinella granulosa and Physalaemus biligonigerus, with an isometric and positive growth of muscles and bones, and most tendons being isometric. This trend represents a synchronic growth of all structures, which is an interesting pattern. Overall, the couple antagonist-agonist muscles do not present the same tendency in the surveyed species. For example, the triceps generally grows with positive allometry, and the coracoradialis (biceps) with negative allometry. On the contrary, in the forearm of Rhinella and Physalaemus, agonist and antagonist muscles presented the same trend. Taken together, these data might indicate that the differential growth of the musculoskeletal-tendinous system might be related to the intrinsic nature of each tissue e.g., tendons increase their lengths by apposition of collagen fibril segments in a complex and hierarchical process (Zhang et al., 2005), meanwhile bone growth in length is mainly achieved through the action of chondrocytes in the proliferative and hypertrophic zones Rauch, 2005).

Manzano et al. (2013) stressed that the tendon is the last tissue to form during limb ontogeny in anurans and that it needs a fully functional limb to reach complete maturity. Our data shows that this delay in tendon appearance and growth occurs after reaching the fully functional stage, as evidenced by its negative allometric growth. On the contrary, muscles and bones show the same positive allometric trend. This coordinated growth trend contrasts with the previous morphogenetic processes in which muscular differentiation seems to be extremely fast compared to the differentiation of the limb skeletal element (Manzano et al., 2013). Our results on postnatal growth are concordant with the observations made by Huang et al. (2015) in mouse mutants (Splotch delayed (Spd) mice (Vogan et al., 1993)). These authors found that the first stage of tendon development—in which muscles span the zeugopodium anchor to autopodium induced tendons—might be better described through positive allometric muscle growth and a negative allometric tendon elongation, and a subsequent reversal of this trend. Thus, the correct assembly of the musculoskeletal-tendinous complex of a limb segment as unity is regulated by differential growth, in a similar way to that proposed by Eilam (1997). In that study, a heterochronic process was suggested as the critical factor to explain body morphology divergence in several rodent taxa.

Heterochrony is a central process driving morphological diversity in mammals (Ravosa, Meyers & Glander, 1993; Maunz & German, 1997; Richardson et al., 2009), which also seems to modulate musculoskeletal-tendinous growth in anurans. The synchronic muscle-bone growth combined with a negative allometric growth of tendon length results in a segment highly occupied by muscle fibers. On the contrary, a positive allometric growth of the tendons length combined with a synchronic and negative muscle-bone growth would result in a segment highly occupied by tendons. This process could explain, for example, the differences between tendon length of the forearm of a bat or a horse and that of a rat, and would provide a simple mechanism to account for their highly specialized locomotor types.

When the relative growth of the arm and forearm structures of the analyzed anuran species are compared, interesting trends emerge. In the forearm, there is a general trend of a positive bone and muscle allometry, including extensors and flexors, and the already reported delayed tendon growth. In the arm, there is a trend to a positive humerus and triceps allometry, combined with a negative coracoradialis allometry. Strikingly, the coracoradialis tendon presents isometric growth. The described growth of the coracoradialis tendon, combined with negative allometric muscle growth, indicates the presence of a forearm flexor layer with long tendons. Interestingly, the pattern highlighted by Bobbert (2001) as an intriguing design aspect of the human musculoskeletal system (distal muscle–tendon complexes spanning the distance between origin and insertion, with long tendons and very short muscle fibers) was only recorded for the coracoradialis. The longer tendon compared with the muscle length indicates a segment with less force but faster reaction, which could compensate for the great force with slower reaction indicated by the relative growth of the triceps.

In conclusion, our data indicate that the musculoskeletal-tendinous growth is different than posed in our initial hypothesis: limb bones and muscles tend to develop synchronically, with tendons exhibiting a delayed growth.

Supplemental Information

Table S1 The raw of muscles, bones and body measurements and a number of individuals used per species. The specimens were staged in three estimated categories according to Gosner (1960): metamorphic 46 (2); juvenile (3); and adult (4)

Click here for additional data file.

Table S2 Summary of results of multivariate allometry in Rhinella arenarum

Click here for additional data file.

Table S3 Summary of results of multivariate allometry in Rhinella granulosa

Click here for additional data file.

Table S4 Summary of results of multivariate allometry in Trachycephalus typhonius

Click here for additional data file.

Table S5 Summary of results of multivariate allometry in Physalaemus biligonigerus

Click here for additional data file.

Table S6 Summary of results of multivariate allometry in Leptodactylus chaquensis

Click here for additional data file.

Table S7 Summary of results of multivariate allometry in Pleurodema borelli

Click here for additional data file.

We are very thankful to Marta Cánepa and Sonia Kretzschmar from Herpetology Collection of the Fundación Miguel Lillo and Museo de Ciencias Naturales de la Universidad Nacional de Salta for allowing access to herpetological collections. We thank to Sofía Nanni for providing suggestions that greatly improved the manuscript.

Additional Information and Declarations

Competing Interests

Author Contributions

Data Availability

Virginia Abdala is an Academic Editor for PeerJ.

Mónica Soliz conceived and designed the experiments, performed the experiments, analyzed the data, prepared figures and/or tables, authored or reviewed drafts of the paper, and approved the final draft.

María Jose Tulli conceived and designed the experiments, analyzed the data, prepared figures and/or tables, authored or reviewed drafts of the paper, and approved the final draft.

Virginia Abdala conceived and designed the experiments, authored or reviewed drafts of the paper, and approved the final draft.

The following information was supplied regarding data availability:

The raw muscles, bones and body measurements and the number of individuals used per species are available in Table S1.

Tables S2–Table S7 contain a summary of results of multivariate allometry untrimmed (m = 0) as well as trimmed (m = 1) calculations of confidence intervals, opting for the results with lower average standard deviation or bias, trimmed with CI 95% which combines the conservative safety of interval and estimated bias in these analyses.

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
