# Peer review of "Forelimb musculoskeletal-tendinous growth in frogs"

_PeerJ, doi:10.7717/peerj.8618_

## Round 0.1 · original submission · Major Revisions

Both reviewers agree that your work is of good scientific quality.They also proposing several ways to ameliorate the manuscript, including changing Figure 1 with amelioration, and presenting the introduction in a more question-based manner before submitting your revision.

Reviewer 1 ·

Basic reporting

Manuscript #40988 “Forelimb musculoskeletal-tendinous growth in frogs” dealing with the musculoskeletal-tendinous system variation in frogs through ontogenetic perspective while taking into account potential impact of species assignment and locomotor mode.
Manuscript is clear with technically and grammatically correct English and well structured. Introduction section is relevant to the topic with appropriate literature cited. Hypothesis is well defined and relevant as there are only few studies dealing with the ontogenetic variation of the musculotendinous system. Raw data shared.

Experimental design

Research is within aims and scope of the journal and fills knowledge gaps in the fields of anatomy, morphology, evolution and development of anurans. Analyses were done on the seventy-seven specimens belonging to seven anuran species. Sample size is rather small for some species (e.g. Xenopus laevis) with only one or two specimens from specific age category. However, robust and rigorous statistical procedures with the resampling give results that are reliable. Material and methods are clear, easy to follow and easy to replicate.

Validity of the findings

Results and conclusions contribute to the better understanding morphological variation, development and evolution of musculotendinous system in anurans.

Additional comments

Specific comments:
Lines 3, 5, and 7. Replace a,b,c with 1,2,3
Line 142. Erase “Descriptive statistics and” as in Supplementary file only individual values are given.
Line 162. Erase “)” from the end of the line
Suggestion. In order to make results more transparent add table with allometry coefficients and CI intervals as supplementary file (similar to Table 3 but with calculated numerical values instead of +,-,=).

Reviewer 2 ·

Basic reporting

Clear paper

References sholud be revisited

Good structure of the paper. But Figure 1 must be revisited and a Table with "ecological requirements" of the studied species should be useful

Descriptive paper. Please indicate the tested hypothesis into the last paragraph of the introduction

Experimental design

OK. But only one varialbe (length used).

The study is rigorous and stats seem to be very good.

Validity of the findings

In my opinion, this paper can be published in a more anatomical or herpetological journal. Adding Table with locomotor modes, testing the modes (indicating the results) should be useful.

Additional comments

The paper present nice results. But some points must be clarified or emphasized:
- Hypothesis
- Selection of the species and thier modes of locomotion
- Figure 1 should be revisited

Annotated reviews are not available for download in order to protect the identity of reviewers who chose to remain anonymous.

---

## Round 0.2 · accepted · Accept

The new review shows that you revised your manuscript according to reviewer's previous comment so I am please to accept this interesting paper.

Reviewer 1 ·

Basic reporting

Article is written in clear English language with appropriate literature referenced. Introduction section was corrected according to reviewer suggestions. Additional tables were added as a supplementary files upon reviewer request that make results more transparent. Hypothesis were clarified upon the request of reviewer which improved clarity of manuscript.

Experimental design

Experimental design is clear and all analyses are rigorously performed.

Validity of the findings

Conclusion section is corrected according to the reviewer suggestions, supports the results and its easy to follow.